# Toward Greater Autonomy in Materials Discovery Agents: Unifying Planning, Physics, and Scientists

**Lianhao Zhou**[1*]  **Hongyi Ling**[1*]  **Keqiang Yan**[1*]  **Kaiji Zhao**[2]
**Xiaoning Qian**[1,3,4]  **Raymundo Arróyave**[2]  **Xiaofeng Qian**[2,3,5]  **Shuiwang Ji**[1†]

[1]Department of Computer Science and Engineering, Texas A&M University
[2]Department of Materials Science and Engineering, Texas A&M University
[3]Department of Electrical and Computer Engineering, Texas A&M University
[4]Computing and Data Sciences, Brookhaven National Laboratory
[5]Department of Physics and Astronomy, Texas A&M University

**Reviewed on OpenReview:** https://openreview.net/forum?id=Cwq1U8tbWW

## Abstract

We aim at designing language agents with greater autonomy for crystal materials discovery. While most existing studies restrict the agents to perform specific tasks within predefined workflows, we aim to automate workflow planning given high-level goals and scientist intuition. To this end, we propose Materials Agents unifying Planning, Physics, and Scientists, known as MAPPS. MAPPS consists of a Workflow Planner, a Tool Code Generator, and a Scientific Mediator. The Workflow Planner uses large language models (LLMs) to generate structured and multi-step workflows. The Tool Code Generator synthesizes executable Python code for various tasks, including invoking a force field foundation model that encodes physics. The Scientific Mediator coordinates communications, facilitates scientist feedback, and ensures robustness through error reflection and recovery. By unifying planning, physics, and scientists, MAPPS enables flexible and reliable materials discovery with greater autonomy, achieving a 24.9% stable–unique–novel (S.U.N.) rate on MP-20, approximately five times the 4.92% rate of the strongest reported generative baseline, while improving the DFT stability rate from 17.8% to 34.3%. We provide extensive experiments, component ablations, fixed-workflow baselines, and cost analyses across diverse tasks to show that MAPPS is a promising Level 2 human-guided framework for materials discovery.

## 1 Introduction

Materials discovery has significant societal impacts across energy, environment, health, and beyond. However, its current pace remains limited by a heavy reliance on trial-and-error wet-lab experiments. Computational methods, including those based on density functional theory (DFT), have substantially accelerated materials discovery over the past several decades. Nevertheless, DFT calculations remain computationally expensive, making exhaustive exploration of the vast space of candidate materials infeasible. In the past few years, advances in AI for science have led to a new paradigm for scientific discovery by providing significant speed-ups over traditional DFT based methods. There are predictive models (Choudhary et al., 2020; Yan et al., 2022; 2024c;a; Choudhary et al., 2024) proposed to predict physical properties of atomistic systems with remarkable efficiency and accuracy, and generative models (Jiao et al., 2023; Antunes et al., 2024; Yan et al., 2024b; Zhang et al., 2023) that can generate novel and stable materials with desired properties. Powered by large language models (LLMs), recent studies have also started exploring how LLM-based AI agents can support autonomous materials discovery (Jia et al., 2024; Zhang et al., 2024). Currently, most existing studies constrain LLM

---

[*]Equal contribution
[†]Correspondence to: Shuiwang Ji, `sji@tamu.edu`

agents to perform predefined actions specified by human experts, with fixed tasks at each step. In these setups, a fixed discovery pipeline is defined in advance, and LLM agents primarily serve to coordinate AI tools (Zou et al., 2025).

Here, we attempt to enable more autonomy in materials discovery agents by making use of the planning capabilities of LLMs. While LLMs' planning capabilities for generic and complex tasks are unclear and still a topic of intensive research and discussions, we attempt to explore their performance in a constrained setting of materials discovery and particularly in scientific workflow planning. This refers to the construction and adaptation of structured sequences of domain-specific actions designed to solve scientific problems. We show that, while the broader capabilities of LLMs in general-purpose planning remain limited, their emerging ability to perform scientific workflow planning is promising, especially when coupled with human scientist interactions. This focused planning capability opens the door to more autonomous and adaptive agents, enabling systems that can reason about goals, generate workflows, and revise their plans dynamically to achieve scientific objectives.

Concretely, unlike most existing approaches that constrain agents with predefined workflows tailored to specific tasks, our focus is on enabling agents to plan workflows and reason independently. Instead of prescribing step-by-step procedures, we provide only high-level goals and scientific intuition, allowing agents to determine the sequence of actions required to achieve discovery objectives. In addition, we design our agent system to be physics-informed and include human experts in the loop. This setup enriches the agent's scientific knowledge beyond textual data, mitigates risk, and allows expert guidance to influence the agent's decisions. To this end, we propose Materials Agents unifying planning, Physics, and Scientists, known as MAPPS, a multi-agent system equipped with a Workflow Planner, a Tool Code Generator, and a Scientific Mediator. These three agents, coupled with human scientists, collaboratively drive materials discovery by planning tasks, generating code, and integrating expert guidance. On MP-20, MAPPS achieves a 24.9% S.U.N. rate, approximately five times the 4.92% rate of the strongest reported generative baseline, while its DFT stability rate improves from 17.8% to 34.3%. We further evaluate fixed non-agent workflows, component ablations, the role of human guidance, and computational cost. These results position MAPPS as a Level 2 human-guided planning system rather than a fully autonomous discovery system.

## 2 Materials Agents Unifying Planning, Physics, and Scientists

The goal of materials discovery is to discover novel materials structures with desirable physical or chemical properties. Following Yan et al. (2024b), we represent each crystal structure as a tuple $\mathbf{M} = (\mathbf{X}, \mathbf{P}, \mathbf{L})$, where $\mathbf{X} = [\mathbf{x}_1, \mathbf{x}_2, \cdots, \mathbf{x}_n] \in \mathbb{R}^{d_x \times n}$ denotes the list of $n$ one-hot representations of atom types in the unit cell, $\mathbf{P} = [\mathbf{p}_1, \mathbf{p}_2, \cdots, \mathbf{p}_n] \in \mathbb{R}^{3 \times n}$ represents the Cartesian coordinates of the atoms, and $\mathbf{L} = [\boldsymbol{\ell}_1, \boldsymbol{\ell}_2, \boldsymbol{\ell}_3] \in \mathbb{R}^{3 \times 3}$ specifies the lattice matrix containing three basic vectors to describe periodic boundary of the unit cell.

In this paper, we focus on three types of tasks, including crystal generation, crystal structure prediction, and property-guided generation. Crystal generation is the unconditional generation of stable crystal structures without predefined constraints. Crystal structure prediction aims to generate a stable structure given a specific chemical composition. Property-guided generation seeks to design crystal structures that satisfy desired property criteria, such as a target band gap or formation energy. Rather than generating the final structure in a single step, these tasks can be formulated as sequential decision-making problems, where an agent constructs the crystal structure $\mathbf{M}$ through a series of $T$ actions $(a_1, a_2, \ldots, a_T)$. The process may start from an empty structure or from a candidate retrieved from a database, followed by iterative refinement to achieve the design objective.

### 2.1 Different Levels of Autonomy in Science Agent Design

Given the complexity of such tasks, it becomes crucial to understand the level of autonomy given to the agents performing them. We define three levels of autonomy for agents in materials science discovery, characterized by the agent's freedom in planning workflows.

**Level 1 − Tool-Executing Agents.** The agent performs specific tasks within a fixed, human-designed workflow. It operates as a tool integrator or step executor, typically relying on predefined templates or

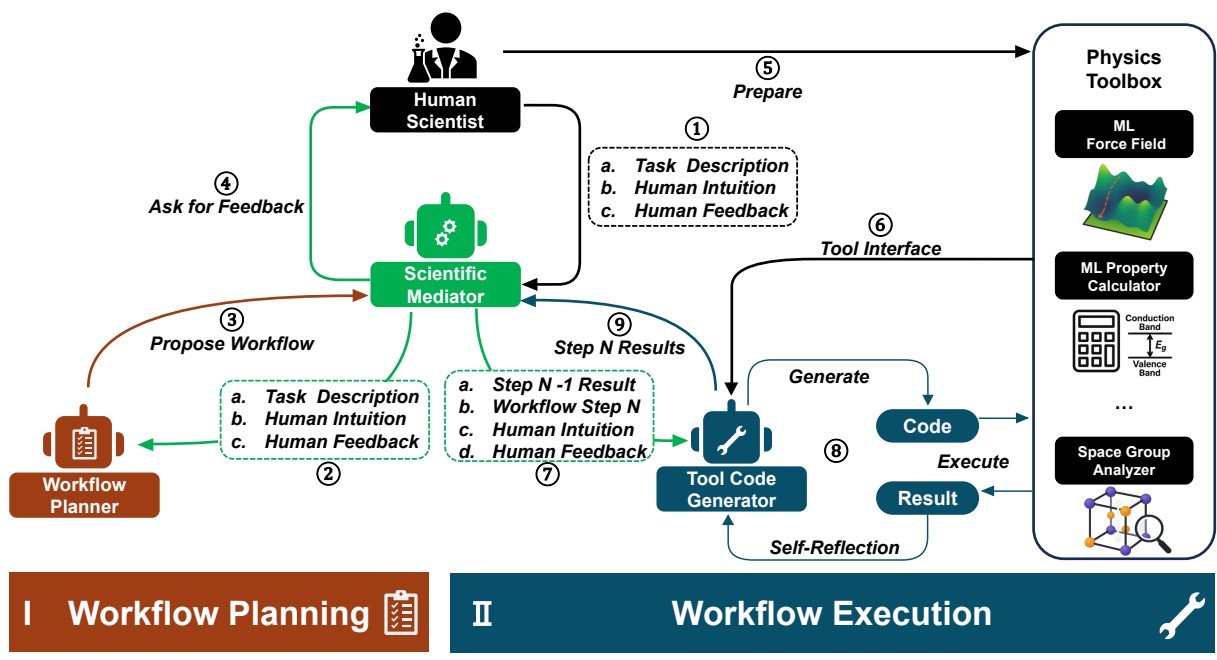

Figure 1: MAPPS Agent Framework. The MAPPS framework consists of three key modules: the Workflow Planner, Tool Code Generator, and Scientific Mediator, which collaboratively drive structure discovery by planning tasks, generating code, and integrating expert guidance. As shown in the figure, the process begins with the human scientist providing a task description, domain intuition, and optional feedback. The Scientific Mediator interprets these inputs and passes them to the Workflow Planner, which proposes a multi-step workflow tailored to the scientific objective. Once the workflow is approved by the human, the Scientific Mediator forwards it to the Tool Code Generator, which translates each workflow step into executable code. This module invokes domain-specific tools from the Physics Toolbox, such as ML Force Fields for structure relaxation, ML Property Calculators for evaluating physical properties, and Space Group Analyzers for symmetry analysis. After execution, results are returned, and the system can optionally engage in self-reflection to detect and correct errors, iteratively improving generated code.

direct calls to existing tools, such as running a DFT calculation. In this setup, planning freedom is minimal, since agents do not alter the overall workflow or sequence of operations and merely automate individual components.

**Level 2 – Human-Guided Planning Agents.** The agent proposes workflows by itself, but with human-provided intuition, constraints, or intermediate goals. For instance, the agent may decompose a complex task into sub-tasks based on the domain knowledge provided by experts. Human feedback or verification helps prune the workflow space, allowing for more flexibility than Level 1 while maintaining scientific plausibility and feasibility.

**Level 3 – Fully Autonomous Planning Agents.** The agent has full freedom to design and adapt workflows from scratch, with no predefined sequence or human-imposed constraints. It decides which tools to use, how to combine them, and in what order. While this level enables the highest flexibility and potential for novel discoveries, it poses significant challenges in ensuring workflow validity, reliability, and scientific correctness.

## 2.2 Overview of MAPPS

Most existing agent methods in materials science use LLMs as Level 1 autonomous agents, where the model executes isolated tasks within human-curated workflows or serves as a natural language interface to domain-specific tools. These approaches treat the LLM primarily as a tool user, relying heavily on fixed, expert-designed procedures. While effective in individual components, such Level 1 agents are constrained in their ability to adapt or generalize to new scientific challenges.

To improve the autonomy of LLM agents for science, we aim to move beyond systems where agents act solely as tool executors within fixed, human-designed workflows. Instead, we propose a multi-agent framework MAPPS that achieves Level 2 autonomy by enabling agents to actively design and follow their own workflows. Instead of following predefined steps, the agents construct sequences of actions to solve scientific problems, guided by high-level human input such as goals, constraints, or domain heuristics. This design allows the agents to independently develop both solutions and the necessary tools, leading to better adaptability and scientific creativity.

Specifically, our multi-agent framework integrates three core components, including **Workflow Planner**, **Tool Code Generator**, and **Scientific Mediator**. The Workflow Planner uses an LLM to decompose high-level scientific goals into adaptive, multi-step plans. The Tool Code Generator synthesizes executable code for each step and incorporates physics-based tools, ensuring that outputs are grounded in fundamental physical laws. The Scientific Mediator coordinates communication between agents and humans, maintaining consistency and tracking progress. See Figure 1 for an overview of our MAPPS agent framework.

## 2.3 Workflow Planner

To enable autonomous scientific planning, the Workflow Planner is built on a large reasoning model (LRM), which is an advanced LLM with enhanced planning and reasoning capabilities, to generate workflows for different tasks. Let $\tau \in \mathcal{T}$ denote a high-level task description, where $\mathcal{T}$ is the space of natural language prompts that specify scientific goals. Given a high-level task description $\tau$, the goal of the Workflow Planner is to generate a workflow consisting of $T$ actionable steps, $\mathbf{A} = (a_1, a_2, \ldots, a_T)$, where each $a_t \in \mathcal{A}$ corresponds to a structured scientific or data processing action, and $\mathcal{A}$ is the space of all executable operations. Formally, the workflow generation process can be described as

$$\mathbf{A} \sim P_\theta(\mathbf{A} \mid \tau) = \prod_{t=1}^{T} P_\theta(a_t \mid a_{<t}, \tau), \tag{1}$$

where $P_\theta$ is parameterized by the pretrained LRM and $a_{<t} = (a_1, \ldots, a_{t-1})$ denotes the sequence of previously generated steps. However, as shown in Section 4.5, relying solely on a high-level task description $\tau$ often results in invalid or impractical workflows due to the model's limited domain knowledge. To address this, we introduce an auxiliary input $\iota \in \mathcal{I}$, where $\mathcal{I}$ denotes the space of human-provided intuition, such as domain-specific heuristics. Formally, the refined generation process can be expressed as

$$\mathbf{A} \sim P_\theta(\mathbf{A} \mid \tau, \iota) = \prod_{t=1}^{T} P_\theta(a_t \mid a_{<t}, \tau, \iota). \tag{2}$$

This refinement guides the model toward more valid plans, while slightly sacrificing the agent's freedom in exploring arbitrary planning strategies.

In our implementation, human intuition $\iota$ is provided in a structured prompt with the task description $\tau$. This includes relevant constraints, known physical principles, or useful heuristics, leading to excluding unfeasible operations such as environment setup or model loading from the action space $\mathcal{A}$. Note that the LRM is instructed to output a well-structured workflow containing at most $T = 5$ steps. By injecting expert intuition into the generation process, the Workflow Planner supports Level 2 autonomy, allowing agents to plan more effectively while retaining a human-in-the-loop safeguard. The following template and example demonstrate how human intuition shapes the generated workflow. See Appendix A.1 for the detailed workflow and the prompt template used by the Workflow Planner.

### 2.4 Scientific Mediator

To support structured interaction between human scientists and autonomous agents, we introduce the Scientific Mediator, a central coordination module that enables a lightweight human-in-the-loop mechanism. While the system is capable of autonomous operation, the Scientific Mediator incorporates human guidance at key decision points to ensure scientific reliability and task relevance.

The process begins when the scientist provides a high-level task description $\tau$. The Scientific Mediator forwards $\tau$ to the Workflow Planner, which generates a structured, multi-step workflow $\mathbf{A} = (a_1, a_2, \ldots, a_T)$. The mediator is not a passive context concatenator: before execution, it audits the proposed workflow for missing physical constraints, ambiguous operations, and unavailable tools, and returns incomplete plans for revision. Once the scientist approves the plan, the mediator initiates execution step by step. At each step $t$, it constructs an augmented input context $\xi_t = (a_t, r_{t-1}, \iota_t)$, where $a_t$ is the current action, $r_{t-1}$ denotes the intermediate result from the previous step, and $\iota_t$ is the human intuition at step $t$. During execution, the mediator acts as an error gatekeeper: it captures syntax and runtime tracebacks, routes them to the Tool Code Generator for self-correction, and escalates only persistent failures or physical ambiguities to the scientist. This context is sent to the Tool Code Generator to synthesize executable code to perform action $a_t$ and compute the corresponding results $r_t$. Before proceeding to step $t + 1$, the Scientific Mediator queries the human for approval or feedback on $r_t$, enabling intervention when necessary. This iterative process maintains a human-in-the-loop mechanism while preserving the autonomy of the system. In this way, the Scientific Mediator plays a crucial role in bridging autonomous agents with human experts, ensuring adaptability and scientific validity.

### 2.5 Tool Code Generator

Given the workflow $\mathbf{A}$ generated by the Workflow Planner, the Tool Code Generator is an autonomous agent responsible for translating each workflow step into executable Python functions. Specifically, at each step $t$, it receives the input context $\xi_t = (a_t, r_{t-1}, \iota_t)$ from the Scientific Mediator and synthesizes a Python function to perform action $a_t$ and compute the result $r_t$.

In addition, the Tool Code Generator operates with a set of domain-specific physics tools denoted as $\Psi = \{\psi_1, \psi_2, \ldots, \psi_n\}$, where each $\psi_i$ represents an individual physics tool. The Tool Code Generator uses a pretrained LRM to generate executable code, integrating these domain-specific physics tools to ensure physically grounded outcomes. For instance, within a crystal structure prediction workflow, the Tool Code Generator integrates a space group analyzer in the initial step to validate symmetry preservation from prototype structures. Subsequently, ML Force Fields (MLFFs) are used to efficiently relax candidate structures towards energetically favorable configurations, significantly reducing computational overhead compared to DFT calculations.

To enhance robustness, the Tool Code Generator includes a self-reflection mechanism. If execution of the generated code results in runtime errors, a diagnostic error signal $e_t$ is generated, triggering the Tool Code Generator itself to revise and regenerate the code. Formally, the initial code generation and self-reflection-based revision processes can be expressed as

$$c_t \sim P_\phi(c_t \mid \xi_t, \Psi) = P_\phi(c_t \mid a_t, r_{t-1}, \iota_t, \Psi), \tag{3}$$

where $c_t$ represents the code generated for executing step $a_t$. The revision upon encountering an error is formulated as

$$c_t' \sim P_\phi(c_t' \mid \xi_t, e_t, \Psi), \tag{4}$$

where $c_t'$ is the revised code generated for executing step $a_t$. The result $r_t$ is computed by executing the generated code $c_t$ if no error occurs; otherwise, it is obtained by executing the revised code $c_t'$. See Appendix A.2 for the detailed tool code generation process.

## 3 Related Work

**Crystal Structure Generation.** Existing methods for 3D crystal generation can be broadly categorized into diffusion-based generative models and language model-based approaches. CDVAE (Xie et al., 2022) models the generation of crystal structures by combining variational autoencoders with denoising diffusion. It learns a latent representation of crystals and gradually refines noisy samples into valid structures through a learned reverse diffusion process. DiffCSP (Jiao et al., 2023) is a diffusion-based approach specifically designed for crystal structure prediction. It conditions the generation on a given chemical composition and guides the denoising process using surrogate energy models to produce low-energy, stable structures. Mat2Seq (Yan et al., 2024b) is a language-model-based approach that converts crystal structures into token sequences and trains an autoregressive transformer to generate them. It converts 3D crystal structures into invariant and complete 1D sequences that language models can take as input. CrystaLLM (Antunes et al., 2024) trains a language model for crystal generation using text-like representations of crystal structures, i.e., CIF files. However, these generative approaches are designed with specific tasks. They serve as tools that assist in discovery but cannot plan, reason, guide, or control the discovery process.

**LLM Agents for Science.** LLM agents are now widely adopted across various scientific domains. In the materials science domain, several systems have been developed to use LLMs for autonomous material discovery. AtomAgents (Ghafarollahi & Buehler, 2025) uses a multi-agent framework combining physics-based simulations and multi-modal data integration to design and discover new alloys. OSDA Agent (Hu et al., 2025) focuses on zeolite synthesis by integrating molecule generation, quantum evaluations, and reflective feedback to identify suitable organic structure directing agents. LLMatDesign (Jia et al., 2024) uses LLMs to translate human instructions into material modifications, applying iterative updates to optimize properties. MatLLMSearch (Gan et al., 2025) demonstrates that pre-trained LLMs, combined with evolutionary search algorithms, can generate stable crystal structures without additional fine-tuning. Similarly, in other scientific fields, LLM agents have been developed to integrate domain-specific tools within structured workflows (Ghafarollahi & Buehler, 2024; Liu et al., 2024b; Kang & Kim, 2024; Liu et al., 2024a; Huang et al., 2024). Systems such as ChemCrow (Bran et al., 2024) enable autonomous chemical synthesis by combining LLMs with several chemistry tools. Although these systems are promising, LLMs are typically used as tool users who execute predefined steps in workflows designed by human experts and depend heavily on existing domain tools and infrastructure in these systems. As we discussed in Section 2.1, this Level 1 usage pattern constrains the autonomy and adaptability of the agent in complex science discovery tasks.

**Differences with Prior Work**. MAPPS distinguishes itself from prior agent systems through its ability to autonomously design workflows, implement code, and incorporate intuition and feedback from human experts. We go beyond Level 1 tool-executing agent systems by introducing a Level 2 framework, where agents perform human-guided planning rather than merely executing predefined tasks. Through extensive experiments, we demonstrate that the MAPPS system outperforms Level 1 agent systems, even when those systems follow workflows carefully crafted by human experts.

## 4 Experiments

In this section, we evaluate MAPPS on a diverse range of real-world material discovery tasks, including crystal structure generation, crystal structure prediction, and discovering crystal structures with desired properties. The experimental results demonstrate that our proposed multi-agent framework is able to complete these challenging tasks. Additionally, we present a study in Section 4.5 to analyze the workflow generations. We conduct our experiments using OpenAI API and a single NVIDIA A100 GPU.

### 4.1 Crystal Structure Generation

**Setup.** A major goal of materials science is to discover stable and novel crystals. We first evaluate the ability of our proposed multi-agent framework to generate stable crystal structures. We consider two datasets, including MP-20 (Jain et al., 2013) and Matbench (Dunn et al., 2020). MP-20 includes 45,231 stable materials from the Materials Project, covering materials with a maximum of 20 atoms per unit cell and within 0.08 eV/atom of the convex hull. We follow (Gan et al., 2025) to process Matbench. The datasets are used as

Table 1: Results for crystal structure generation on the MP-20 dataset.

| Model | Validity Rate (%) | | Metastability Rate(%) | Stability Rate (DFT) (%) | S.U.N Rate (DFT)(%) |
|---|---|---|---|---|---|
| | Structural | Composition | M3GNet ($E_{hull} < 0.1$) | | |
| CDVAE | 100 | 86.7 | 28.8 | 1.6 | 1.43 |
| DiffCSP | 100 | 83.3 | – | 5.1 | 3.34 |
| FlowMM | 96.9 | 83.2 | – | 4.7 | 2.34 |
| CrystalTextLLM | 99.6 | **95.4** | 49.8 | 5.3 | – |
| FlowLLM | 99.9 | 90.8 | – | 17.8 | 4.92 |
| MAPPS | **100** | 94.0 | **95.0** | **34.3** | **24.9** |

Table 2: Large-scale crystal structure generation results on MP-20. All generated candidates are evaluated without pre-screening.

| Method | Number of candidates | DFT stability (%) | S.U.N. (%) |
|---|---|---|---|
| MAPPS | 1,000 | 34.3 | 24.9 |
| MAPPS | 10,000 | 28.8 | 19.2 |

the retrieval database of our method and the training set of the baselines. We generate 1,000 candidates on MP-20 and Matbench and evaluate every generated candidate without pre-screening. To assess scalability, we additionally generate 10,000 candidate structures on MP-20 using the same prototype database, generation rules, physical validation criteria, and uniqueness and novelty protocols as in the 1,000-candidate experiment. All 10,000 candidates are included in the evaluation without pre-screening or selective reporting. We evaluate the quality of generated crystal structures using four metrics: validity rate, metastability, stability, and S.U.N rate. Following Xie et al. (2022); Court et al. (2020); Miller et al. (2024), we compute both structural and compositional validity percentages based on heuristic checks of interatomic distances and charge balance, respectively. For metastability, we adopt the approach of Gan et al. (2025), using CHGNet (Deng et al., 2023) and M3GNet (Chen & Ong, 2022) as surrogate models to estimate the fraction of structures with decomposition energies below thresholds of 0.1 eV/atom and 0.03 eV/atom. Stability is further assessed through DFT calculations, where a structure is considered stable if its energy above the convex hull ($E_{hull}$) is less than 0 eV/atom. Finally, the S.U.N. rate measures the proportion of structures that are stable, unique, and novel.

**Baselines.** We compare MAPPS with the following baseline methods, including (1) CDVAE (Xie et al., 2022), a crystal diffusion variational autoencoder that learns to denoise atomic coordinates and atom types through a diffusion process; (2) DiffCSP (Jiao et al., 2023), which is a diffusion-based generative model that uses a periodic E(3)-equivariant network to jointly generate lattice parameters and fractional atomic coordinates, ensuring symmetry-aware crystal generation; (3) FlowMM (Miller et al., 2024), a Riemannian flow matching model tailored to crystal symmetries, offering efficient and accurate generation of periodic structures; (4) CrystalTextLLM (Gruver et al., 2024), which leverages fine-tuned large language models to generate crystal structures from string-based representations, supporting both unconditional and text-guided generation; (5) FlowLLM (Sriram et al., 2024), which fine-tunes an LLM to learn an effective base distribution of meta-stable crystals in a text representation.and (6) MatLLMSearch (Gan et al., 2025), which integrates pre-trained LLMs with evolutionary search to iteratively generate and optimize crystal candidates based on structural and property constraints.

**Results.** The results in Table 1 show that MAPPS outperforms several generative model-based baselines, including CDVAE, DiffCSP, FlowMM, and CrystalTextLLM, on the MP-20 dataset. Notably, our approach does not require training any new model. Instead, it successfully extracts the scientific knowledge embedded in pretrained LLMs to solve the crystal structure generation task. This demonstrates that LLMs possess strong capabilities for understanding scientific concepts and facilitating materials discovery. Moreover, the results in Table 3 demonstrate that our MAPPS surpasses MatLLMSearch, a recent baseline that combines LLMs with evolutionary algorithms, on the Matbench dataset. This result highlights that LLMs are not merely useful for replacing individual components in an algorithmic pipeline but can serve as central reasoning engines for end-to-end scientific design. We also provide some examples of generated crystal structures in Figure 2. In the large-scale experiment with 10,000 MP-20 candidates, MAPPS achieves a DFT stability rate

Table 3: Results for crystal structure generation on the Matbench dataset.

| Model | Validity Rate (%) | | Metastability Rate (%) | | |
|---|---|---|---|---|---|
| | Structural | Composition | M3GNet ($E_{hull} < 0.1$) | CHGNet ($E_{hull} < 0.1$) | CHGNet($E_{hull} < 0.03$) |
| MatLLMSearch | 100 | **79.4** | 81.1 | 76.8 | 56.5 |
| MAPPS | **100** | 76.9 | **93.8** | **95.9** | **84.3** |

Figure 2: Examples of crystal structures generated in the crystal structure generation task.

of 28.8% and a S.U.N. rate of 19.2%. Although these values are lower than those of the 1,000-candidate run, they show that MAPPS retains substantial stability, uniqueness, and novelty at a tenfold larger generation scale. Because all generated candidates are evaluated without pre-screening, the result also confirms that the reported performance is not obtained by selecting a small favorable subset from a much larger candidate pool.

## 4.2 Crystal Structure Prediction

While the promising results in Section 4.1 highlight the effectiveness of our proposed framework, we further evaluate its capability on the crystal structure prediction (CSP) task, which involves predicting the stable structure for a given composition. We use the MP-20 and MPTS-52 dataset(Jiao et al., 2023), a challenging benchmark containing $40,476$ structures with up to 52 atoms per unit cell. In addition, we consider the challenge set introduced by Antunes et al. (2024), which focuses on crystals that have only recently been discovered in the literature, to assess MAPPS's ability to uncover novel structures. Two metrics are used to evaluate the quality of generated crystal structures, namely match rate and RMSE. Match rate measures the fraction of generated structures that match the ground-truth structures, as determined by the `StructureMatcher` class in `pymatgen` (Ong et al., 2013). RMSE measures the structural difference between each matched generated structure and its ground-truth counterpart.

**Baselines.** On the MP-20 and MPTS-52 datasets, we compare MAPPS with several baseline approaches, including language model-based methods such as CrystaLLM (Antunes et al., 2024) and Mat2Seq (Yan et al., 2024b), as well as diffusion-based methods such as CDVAE (Xie et al., 2022) and DiffCSP (Jiao et al., 2023). All baselines are trained using the training sets. To ensure a fair comparison, we also provide our agents with access to the corresponding training data for retrieval. For the challenge set, we compare our method with CrystaLLM (Antunes et al., 2024).To ensure a fair comparison, we use the same data, which was used by CrystaLLM for training, as our retrieval source. Additionally, all models are evaluated under the same setting of generating a one-shot candidate structure for each input composition. We also implement two Level 1 non-agent retrieval-and-substitution baselines. The basic baseline retrieves prototypes with identical stoichiometric ratios and element categories. The stronger human-engineered baseline additionally groups elements by chemical family and preserves space-group symmetry during substitution.

**Results.** The results in Table 4 show that our approach achieves the highest match rate and the lowest RMSE on both datasets, indicating superior performance of MAPPS on the crystal structure prediction task. On the MP-20 dataset, our method attains a match rate of 63.9% and an RMSE of 0.022, outperforming all baselines. The performance gains are even more significant on the more challenging MPTS-52 dataset. MAPPS achieves a match rate of 27.6% and an RMSE of 0.097, significantly outperforming the next-best baseline Mat2Seq by 4.5% in match rate. Notably, on the newly introduced challenge set, MAPPS achieves a

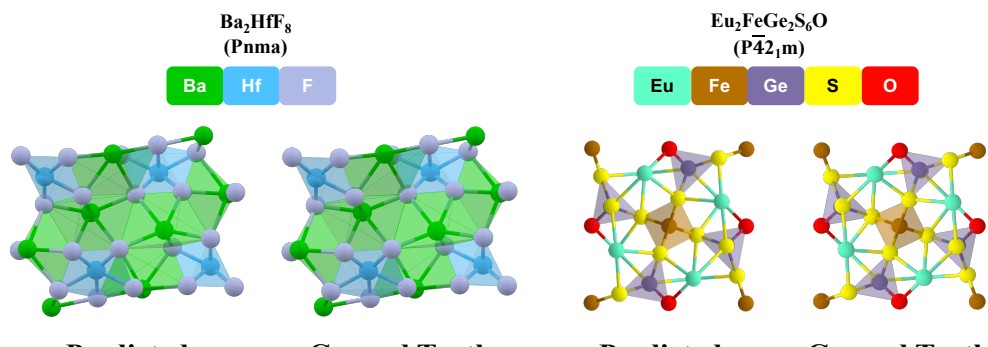

**Figure 3:** Examples of predicted and ground-truth crystal structures in the crystal structure prediction task.

Table 4: Results for crystal structure prediction on three benchmarks, including MP-20, MPTS-52, and the challenge set.

| Model | MP-20 | | MPTS-52 | | Challenge Set | |
|---|---|---|---|---|---|---|
| | Match Rate | RMSE | Match Rate | RMSE | Match Rate | RMSE |
| CDVAE | 33.9% | 0.105 | 5.34% | 0.211 | – | – |
| DiffCSP | 51.5% | 0.063 | 12.2% | 0.179 | – | – |
| CrystaLLM | 58.7% | 0.041 | 19.2% | 0.111 | 22.4% | 0.090 |
| Mat2Seq | 61.3% | 0.040 | 23.1% | 0.109 | – | – |
| MAPPS | **63.9%** | **0.022** | **27.6%** | **0.097** | **31.0%** | **0.055** |

match rate of 31.0% and an RMSE of 0.055, outperforming CrystaLLM by 8.6% in match rate and showing a substantial improvement. These results underscore the effectiveness of MAPPS, demonstrating that Level 2 autonomous agents can generate high-fidelity materials.

### 4.3 Discovering crystal structures with desired properties

**Setup.** We further evaluate MAPPS's ability to discover crystal structures with desired electronic properties. Specifically, we focus on generating structures with target bandgap values. We consider two distinct settings, including generating crystals with high bandgaps, defined as bandgap values higher than 3 eV, and generating crystals with low bandgaps, defined as bandgap values less than 0.5 eV. For each condition, we generate 500 crystal structures. For retrieval, we use the JARVIS-DFT dataset (Choudhary et al., 2020), which contains 61,541 crystal structures along with their corresponding bandgap values. To evaluate the generated structures at scale, we predict their bandgap values using the pretrained Comformer property predictor and report the percentage that meet the high- and low-bandgap criteria. The DFT-computed bandgap distributions under the two generation settings are shown in Appendix A.5.

**Results.** Table 7 demonstrates that MAPPS can effectively discover crystal structures with target band gap properties. Specifically, under the high band gap setting, 74.6% of the generated crystals have band gap values greater than 3 eV, while under the low band gap setting, 92.2% of the generated crystals have band gap values below 0.5 eV. Moreover, the generated structures show high quality, achieving over 90% validity, uniqueness, and novelty.

### 4.4 Component Attribution and Reproducibility

**Agent core components.** The Tool Code Generator is the execution engine and therefore cannot be removed while retaining an executable system. Its practical effectiveness, however, depends on the surrounding planning and feedback loops. Removing the Scientific Mediator disables workflow auditing and automated traceback routing; minor syntax or runtime errors then either terminate execution or require repeated human debugging. Replacing the Workflow Planner with a rigid prototype-design template yields the performance shown in Table 5. The fixed template cannot dynamically incorporate strict stoichiometric matching, symmetry

Table 5: Comparison with fixed Level 1 workflows and ablations on the crystal structure prediction benchmarks. Each method generates one candidate per target. RMSE is computed on matched structures.

| Method | MP-20 | | MPTS-52 | | Challenge Set | |
|---|---|---|---|---|---|---|
| | Match Rate | RMSE | Match Rate | RMSE | Match Rate | RMSE |
| Basic fixed retrieval + substitution | – | – | – | – | 3.4% | – |
| Strong human-engineered fixed workflow | 59.0% | 0.0208 | 23.7% | 0.0425 | 27.6% | 0.0482 |
| Rigid prototype template (no dynamic planning or later feedback) | 58.4% | 0.0205 | 19.1% | 0.0388 | 22.3% | 0.0425 |
| MAPPS | **63.9%** | 0.0220 | **27.6%** | 0.0970 | **31.0%** | 0.0550 |

Table 6: Effect of optional CHGNet relaxation on MAPPS predictions. The main comparison in Table 4 follows the standard one-shot protocol without downstream relaxation.

| | MP-20 | | MPTS-52 | | Challenge Set | |
|---|---|---|---|---|---|---|
| | Match Rate | RMSE | Match Rate | RMSE | Match Rate | RMSE |
| MAPPS + CHGNet relaxation | 63.30% | 0.017 | 27.10% | 0.033 | 29.31% | 0.037 |

Table 7: Evaluation under Bandgap-Constrained Generation Conditions

| Generation Condition | Condition Satisfaction (Comformer) | Validity | Uniqueness | Novelty |
|---|---|---|---|---|
| Bandgap > 3 eV | 74.6% | 97.8% | 96.4% | 94.8% |
| Bandgap < 0.5 eV | 92.2% | 91.6% | 98.6% | 98.0% |

preservation, charge balance, and task-specific chemical constraints, and consequently underperforms MAPPS across all three CSP benchmarks.

**Human guidance and retrieval.** Human intuition supplies high-level scientific direction rather than step-by-step code. Without this guidance, the planner often proposes impractical actions, such as training a new generative model inside a tool-use environment; the workflow-validity results in Section 4.5 therefore support our Level 2 categorization. Retrieval is also an explicit source of structural priors. A purely blind, non-retrieval search is computationally impractical for the settings studied here. Importantly, retrieval alone does not explain the results: the two fixed retrieval-and-substitution pipelines in Table 5 remain below MAPPS, especially on the Challenge Set.

**Role of physics tools and screening.** In CSP, the main comparison excludes downstream relaxation; optional CHGNet relaxation is reported separately in Table 6 and primarily improves local geometric precision. In CSG, MLFF is used sparsely during test-driven code refinement (20 evaluations) rather than as a post-hoc filter over the final population; all 1,000 generated candidates are evaluated. In CSD, Comformer is used as the task-specific property evaluator and no downstream relaxation is performed.

**Computational and human cost.** For CSP, MAPPS generates one candidate per target, uses 30 LLM API calls (approximately 120,000 tokens), and performs no downstream DFT evaluation. The observed code-bug rate is 8.3%; all syntax and runtime errors are resolved through self-reflection. Human input is requested three times, solely to clarify a file-naming convention, broaden group-level chemical-similarity constraints, and protect symmetry-equivalent sites. For CSG, MAPPS generates and validates 1,000 candidates using 12 LLM API calls, with a 0% code-bug rate and no human intervention. The large-scale physical evaluation comprises 1,000 DFT calculations, preceded by MLFF pre-relaxation. For CSD, MAPPS generates 500 property-targeted candidates using eight LLM API calls and Comformer predictions, with a 0% code-bug rate, no DFT calculations, and no human intervention.

### 4.5 Can current LLMs achieve level 3?

To investigate whether current LLMs can achieve Level 3 autonomy, we evaluated three models—GPT-4o-mini, GPT-4o, and O3-mini—on their ability to design workflows from scratch for three materials discovery tasks: crystal structure prediction (CSP), crystal structure design (CSD), and crystal structure generation (CSG).

Table 8: Validity (%) and workflow length under CSP, CSD, and CSG tasks.

| LLM | Validity W/ Human Intuition | | | Validity W/O Human Intuition | | | Avg Workflow Length |
|---|---|---|---|---|---|---|---|
| | CSP | CSD | CSG | CSP | CSD | CSG | |
| GPT-4o-mini | 0% | 0% | 10% | 0% | 0% | 0% | 5.0 |
| GPT-4o | 10% | 30% | 40% | 0% | 0% | 0% | 5.0 |
| O3-mini | 60% | 60% | 100% | 0% | 0% | 0% | 4.37 |

As shown in Table 8, performance is highly dependent on the model and the provision of human expertise. The advanced reasoning model, O3-mini, significantly outperforms the others when guided by human intuition, achieving validity rates of 60% for CSP and CSD, and 100% for CSG. In contrast, GPT-4o and GPT-4o-mini struggle even with guidance. Critically, without human intuition, all models fail completely, with workflow validity dropping to 0% across all tasks. This highlights that even the most advanced LLMs are not yet capable of unguided, autonomous workflow generation. We also note that less advanced models tend to produce fixed-length, five-step workflows, often including redundant steps, whereas O3-mini generates more concise and effective plans, averaging 4.37 steps.

Furthermore, an attempt to replace the human scientific mediator with a separate LLM agent also proved ineffective, resulting in a 0% workflow validity rate. This aligns with findings that LLMs struggle to reliably evaluate complex reasoning without external grounding. These results collectively underscore that current LLMs cannot achieve true Level 3 autonomy, as they critically depend on human expertise for heuristic guidance, logical correctness, and strategic efficiency.

# 5 Summary and Outlook

We introduce MAPPS, a Level 2 human-guided multi-agent framework that combines planning, physics-based tools, and scientist input for materials discovery. Its Workflow Planner dynamically constructs task-specific workflows, the Tool Code Generator implements them as executable programs, and the Scientific Mediator audits plans, routes execution errors, and selectively requests expert clarification. Across crystal generation, structure prediction, and property-guided discovery, MAPPS outperforms generative and fixed Level 1 baselines. The added ablations show that retrieval and physics tools are useful but do not alone account for the gains: rigid human-engineered workflows underperform dynamic planning, particularly under unfamiliar constraints. MAPPS nevertheless remains dependent on high-level scientific guidance and therefore does not achieve Level 3 autonomy. Other limitations include reliance on external models and databases, the cost of physical validation, and the current focus on crystalline materials. Extending the framework to molecules, polymers, experiments, and more open-ended scientific settings will require stronger grounded reasoning, uncertainty calibration, and validation mechanisms.

# 6 Broader Impact

MAPPS is intended as a workflow-assistance and hypothesis-generation system, not as a replacement for expert scientific validation. A multi-stage pipeline that combines LLM planning, learned force fields, property predictors, DFT, and human feedback can create an appearance of strong validation even when individual approximations or assumptions fail. Over-trust may lead researchers to prioritize physically implausible candidates, overlook model bias inherited from retrieval databases, or misinterpret computed stability as experimental synthesizability. We therefore recommend retaining provenance for retrieved prototypes, generated code, model versions, intermediate calculations, and human interventions; reporting uncertainty and failure cases; and independently validating high-value candidates before experimental use. Human experts remain responsible for selecting objectives, reviewing physical assumptions, and determining whether computational evidence is sufficient for downstream decisions.

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

# A Experimental Details

## A.1 Workflow Planning for Crystal Structure Prediction

---
**Workflow Planner Prompt Template**

You are a Workflow Planner. Based on the task requirements and human expert intuition, provide a workflow as a list of necessary steps. The workflow should contain no more than 5 steps. Each step must involve data processing — steps such as environment setup, loading models, or loading data are not considered complete steps by themselves. End your output with a note for human approval or feedback. Each step should be detailed and written on a new line:

Step 1:
Step 2:
...

Task:"task description"

Human intuition:"scientist knowledge"

---

The workflow example in the box below is generated by the **Workflow Planner**, which maps the task description $\tau$ and human intuition $\iota$ to a multi-step action sequence $\mathbf{A} = (a_1, a_2, \ldots, a_T)$. This process follows Equation 2 in the main text:

$$\mathbf{A} \sim P_\theta(\mathbf{A} \mid \tau, \iota) = \prod_{t=1}^{T} P_\theta(a_t \mid a_{<t}, \tau, \iota). \tag{5}$$

In this example, we provide the following inputs:

**Task Description** ($\tau$): "Please predict the stable structure for $Ba_2Fe_2F_9$."

**Human Intuition** ($\iota$): "1. Recent studies commonly employ Machine Learning force fields as an alternative to Density Functional Theory for optimizing structures and calculating energies. 2. Similar chemical compositions might have similar stable structural prototypes. A dataset is available at {path}, providing various structure prototypes."

The output is the following workflow $\mathbf{A}$:

---
**Workflow Example: Crystal Structure Prediction for $Ba_2Fe_2F_9$**

Step 1: Query the structural database for crystal structures with chemical compositions or reduced formulas similar to $Ba_2Fe_2F_9$ to identify promising structural prototypes.
Step 2: Use the retrieved similar prototypes to generate initial candidate structures specifically for $Ba_2Fe_2F_9$, ensuring a diverse set of likely configurations based on known stable arrangements.
Step 3: Optimize these candidate structures using Machine Learning force fields, ensuring the minimization of energy and refinement of lattice and atomic positions.
Step 4: Calculate the total energies for the optimized candidates and compare their stability; the structure with the lowest energy is identified as the most probable stable configuration.
Step 5: Validate the selected structure by cross-referencing with available experimental or high-accuracy computational data, if available, to confirm its stability and consistency with known behavior for similar compounds.
Please review and provide feedback or suggest revisions to the workflow.

---

## A.2 Tool Code Generation

For each workflow step $a_t \in \mathbf{A}$, the **Scientific Mediator** constructs the input context $\xi_t = (a_t, r_{t-1}, \iota_t)$, where $a_t$ denotes the current step description, $r_{t-1}$ is the result of the previous step, and $\iota_t$ is the domain-specific expert intuition for step $t$. This input context is passed to the Tool Code Generator, along with a collection of physics tools $\Psi = \{\psi_1, \psi_2, \ldots\}$ (e.g., **CHGNetCalculator**, **pymatgen**, **ASE**), which represent the available modeling and simulation environments. The Tool Code Generator synthesizes executable code $c_t$ based on this context, following Equation 3 in the main text:

$$c_t \sim P_\phi(c_t \mid \xi_t, \Psi) = P_\phi(c_t \mid a_t, r_{t-1}, \iota_t, \Psi). \tag{6}$$

Below, we provide a concrete example for Step 3. The prompt template used for the Tool Code Generator is shown below.

---

**Tool Code Generator Prompt Template**

You are a Tool Code Generator. Based on the following information (last step result, current workflow step, and expert intuition), please propose complete and executable Python code. The code must define exactly one unique function named 'stepX' (e.g., step1, step2, etc.). All file paths used in the code must be absolute paths. Save the code result files in a folder and output the folder's path. Do not print any other information. Return the processing result. The function must include a descriptive Python comment at the top explaining its purpose and functionality.

Output a JSON object in the following format:

"function name": "stepX",

"code": "Complete Python code"

---

The structured inputs to the Tool Code Generator are:

**Action Description** ($a_3$): "Optimize candidate structures using Machine Learning force fields, ensuring the minimization of energy and refinement of lattice and atomic positions."

**Previous Step Result** ($r_2$): A folder path `'candidates'` containing candidate structures in 'cif' format.

**Human Intuition** ($\iota_3$): "`CHGNetCalculator` is a machine learning force field that approximates DFT-level accuracy and can be used to efficiently optimize lattice and atomic positions."

**Physics Tool Set** ($\Psi$): Includes `CHGNetCalculator`, `ASE`, `pymatgen`.

The following Python function is synthesized by the Tool Code Generator to implement $a_3$.

The code A.1 represents a successful instantiation of the Tool Code Generator output at Step 3, enabling structure relaxation with physics-informed ML force fields.

Table 9: Execution statistics for the three experimental tasks.

| Task | Candidates | LLM calls | Code bug rate | Human interventions |
|------|------------|-----------|---------------|---------------------|
| CSP | 1 per target | 30 | 8.3% | 3 |
| CSG | 1,000 (+10,000 scale-up) | 12 | 0% | 0 |
| CSD | 500 | 8 | 0% | 0 |

**LLM Individual Summarizer**

```python
def step3(chemical_composition):

    import os
    from ase.io import read, write
    from ase.optimize import BFGS
    from ase.constraints import ExpCellFilter
    from chgnet.model import CHGNetCalculator

    input_dir = 'candidates'
    output_dir = 'optimized_candidates'
    os.makedirs(output_dir, exist_ok=True)

    calc = CHGNetCalculator()

    for filename in os.listdir(input_dir):
        if filename.lower().endswith('cif'):
            file_path = os.path.join(input_dir, filename)
            try:
                atoms = read(file_path)
                atoms.set_calculator(calc)
                ecf = ExpCellFilter(atoms)
                opt = BFGS(ecf, logfile=None)
                opt.run(steps=100)
                write(os.path.join(output_dir, filename), atoms)
            except Exception as e:
                print(f'Error processing {file_path}: {e}')

    return output_dir
```

### A.3 Additional Agent-Designed Workflows

**Crystal Structure Generation.** The generated workflow is: (1) categorize the prototype library by the number of unique elements; (2) identify prototypes with high physical plausibility for accommodating new stable elements; (3) substitute only symmetry-equivalent sites while matching chemical groups and oxidation states and enforcing a Shannon ionic-radius tolerance of 10%; and (4) invoke an MLFF to relax test structures and compute total energy and $E_{\text{hull}}$ during workflow refinement. After the code is validated on a small test set, it is scaled to the complete generation run without selecting only favorable outputs.

**Crystal Structure Discovery.** The generated workflow is: (1) bin the prototype library by electronic bandgap; (2) retrieve prototypes correlated with the target interval; (3) substitute symmetry-equivalent sites using chemically motivated trends, choosing heavier and less electronegative same-group elements to decrease the bandgap and lighter and more electronegative elements to increase it; and (4) call Comformer to evaluate the predicted bandgap of every generated structure.

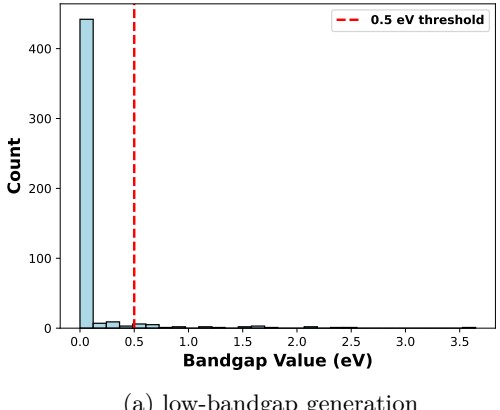 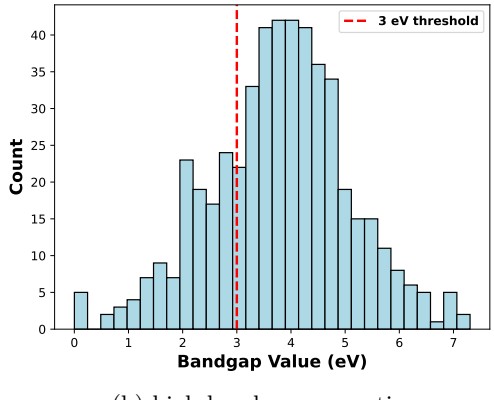

(a) low-bandgap generation          (b) high-bandgap generation

Figure 4: DFT-computed bandgap distributions under two generation settings.

## A.4 Execution Statistics and Human Interventions

As summarized in Table 9, MAPPS requires only limited human intervention across the three tasks. The three interventions in CSP were used to clarify a file-naming convention, expand group-level chemical similarity constraints, and enforce symmetry protection on equivalent sites. They were not requests to repair generated code. All observed syntax and runtime errors were resolved autonomously through the self-reflection loop. CSG and CSD required no human intervention during execution.

## A.5 Bandgap Distribution under Targeted Generation

In Figure 4, we show histograms of DFT-computed bandgaps for structures generated under two targeted conditions in Section 4.3: low-bandgap generation (bandgap < 0.5 eV) and high-bandgap generation (bandgap > 3 eV). Each histogram is computed over 500 generated crystal structures. These plots illustrate how well the generated structures satisfy the intended electronic constraints and how the bandgap distributions differ under each target setting. Under the low-bandgap setting, a significant proportion of the generated structures exhibit bandgap values below the 0.5 eV threshold, demonstrating the model's ability to synthesize narrow-gap materials such as semimetals or small-gap semiconductors. Conversely, in the high-bandgap setting, the bandgap distribution is shifted toward larger values, with many structures achieving bandgaps greater than 3 eV, indicating the successful generation of wide-gap insulating candidates.

## A.6 Evaluation Metric Details

To evaluate the quality of generated crystal structures, we adopt a set of metrics covering structural validity, compositional correctness, thermodynamic stability, uniqueness, novelty, and accuracy of property prediction. Unless otherwise specified, all metrics are computed based on structures post-processed and relaxed by DFT or ML-based surrogates.

**Structural Validity.** A structure is considered structurally valid if all pairwise interatomic distances are greater than or equal to 0.5 Å and the unit cell volume is no less than 0.1 Å$^3$. This ensures that generated structures are physically meaningful and free of atom overlaps or degenerate geometries.

**Compositional Validity.** We assess the physical plausibility of compositions using SMACT Davies et al. (2019), which verifies charge neutrality and electronegativity balance. A crystal is considered compositionally valid if it passes both checks.

**Stability.** We define a crystal as stable if its DFT-calculated energy above the convex hull is below 0.0 eV/atom and it contains at least two unique elements.

**Uniqueness.** To measure diversity, we compute the fraction of stable crystals that are mutually unique. Uniqueness is determined via all-to-all structural comparison using the `StructureMatcher` class from `pymatgen` (Ong et al., 2013). Two crystals are considered duplicates if they match under symmetry-preserving tolerances on lattice, angles, and atomic coordinates.

**Novelty.** A crystal is considered novel if it does not match any existing structure in the original dataset, again based on the `StructureMatcher`. This ensures the generated structures are not trivial rediscoveries.

**Match Rate.** For crystal structure prediction (CSP) tasks, we compute the match rate, defined as the percentage of generated structures that match the ground-truth structure for the given composition, determined using `StructureMatcher`.

**RMSE.** We also report the root mean square error (RMSE) between the fractional coordinates of matching atoms in predicted and true structures, after alignment via symmetry operations and cell transformation. RMSE provides a fine-grained measure of geometric fidelity.

## B    DFT calculations

First-principles density functional theory (DFT) Hohenberg & Kohn (1964); Kohn & Sham (1965) calculations were performed using the Vienna Ab initio Simulation Package (VASP) Kresse & Furthmüller (1996). For stability and S.U.N rate evaluation in Section 4.1, the Perdew-Burke-Ernzerhof (PBE) Perdew et al. (1996) form of the exchange-correlation functional within the generalized gradient approximation (GGA) Langreth & Mehl (1983) was employed. To ensure consistency with the MP-20 dataset, all input settings were generated using the MPRelaxSet class. We determine the DFT energy above hull for the relaxed structures against the Matbench Discovery convex hullRiebesell et al. (2023). The property-guided discovery experiments in Section 4.3 use pretrained Comformer predictions for evaluation and do not perform downstream DFT calculations.

