# OpenReview forum: "Toward Greater Autonomy in Materials Discovery Agents: Unifying Planning, Physics, and Scientists"
_TMLR — Decision pending for TMLR_

### Review · Reviewer_pebh · 2026-04-15

**Summary Of Contributions:**

The paper proposes MAPPS, a new language agent framework for automated crystal material discovery. It consists of a workflow planner, a tool code generator, and a scientific mediator. The core contribution of the paper is that we do not restrict the agent to perform a specific task, and a new LLM also generates high-level goals. Experiment results on crystal structure generation, structure prediction, and structure discovery.

**Additional Comments:**

Here are several questions I would like to ask:
- I cannot grasp the exact function of the scientific mediator. While the authors say that it constructs the augmented input context, it can be obtained just by a naive concatenation. Could you explain more details on the mediator?

- I'm also curios on how much failure occurs during tool code generation and how many times we need to revise the original code. Does it require too long time for handling this?

**Audience:**

Yes

**Audience Explanation:**

Yes, it can be applied to diverse tasks related to crystal structures, ranging from generation, prediction, and discovery.

**Claims And Evidence:**

No

**Claims Explanation:**

While there are strong empirical results, I'm not sure each component of the current framework is necessary as there are lack of ablation studies on each component.

**Requested Changes:**

Here are several changes I would like to suggest:
- Lack of ablation studies: I want to know which component is crucial to the performance
- Detailed experiment setup: Only a single workflow example for structure prediction is provided. I want to know examples of workflows for different tasks, such as generation and discovery.

---

> ### Author Response · Authors · 2026-05-21
>
> We thank the reviewer for the constructive feedback, encouraging comments, and insightful questions. We address each of your comments and requested changes in detail below:
>
> ### 1. Concerning the Exact Function of the Scientific Mediator
> The Scientific Mediator is not a passive context concatenator, but an active Workflow Auditor and Error Gatekeeper designed to minimize human intervention. Before code generation, it audits the Workflow Planner's high-level blueprint to reject incomplete workflows and demand missing details. During execution, it acts as the first line of defense by catching compiler tracebacks and routing them back to the Tool Code Generator for autonomous self-reflection and self-correction, ensuring the system only escalates to a human expert when a fundamental physical ambiguity or persistent failure occurs.
>
> ---
>
> ### 2. Concerning Code Generation Failures and Human Effort
> When utilizing advanced reasoning engines like GPT-5-mini or o3-mini, the reliance on human intervention for code debugging is exceptionally low, as almost all syntax and runtime errors are autonomously resolved via internal self-reflection loops. Under our "Test-Driven Scale-up" paradigm, the agent fully refines its code on a small handful of test cases before deploying it for large-scale production, requiring zero human intervention during the execution of CSG and CSD tasks. For the CSP task, the human was prompted only 3 times by the system to resolve a specific file-naming rule, expand group-level chemical similarity constraints, and enforce symmetry protection on equivalent sites, demonstrating that error handling is highly efficient and requires minimal human time.
>
> ---
>
> ### 3. Concrete Step-by-Step Workflows for CSG and CSD Tasks
> To demonstrate MAPPS’s versatility beyond structure prediction, we outline the concrete, agent-designed workflows for Crystal Structure Generation (CSG) and Crystal Structure Discovery (CSD) below:
>
> #### A. Crystal Structure Generation (CSG) Workflow
> 1. Categorize the reference prototype library based on the number of unique elements in each structure.
> 2. Filter and screen out specific crystal prototypes that possess a higher physical probability of accommodating new, stable elements.
> 3. Perform element substitutions exclusively on equivalent sites using a strict chemical rule-set: matching chemical groups, identical oxidation states, and a Shannon ionic radius tolerance within 10\%.
> 4. Invoke Machine Learning Force Fields (MLFF) to relax the generated structures, dynamically calculating total energy and energy above hull ($E_{\text{hull}}$) to isolate thermodynamically stable crystals.
>
> #### B. Crystal Structure Discovery (CSD) Workflow
> 1. Categorize the reference prototype library into distinct bins based on their electronic bandgap intervals.
> 2. Identify and isolate crystal prototypes that are highly correlated with the user's targeted bandgap range.
> 3. Substitute elements on equivalent sites to dynamically tune electronic properties, choosing heavier, less electronegative elements from the same group to decrease the bandgap, or lighter, more electronegative elements to increase it.
> 4. Interface directly with Comformer to quickly screen and validate the specific bandgap profiles of the newly discovered structures.

---

> ### Author Response · Authors · 2026-05-21
>
> ### 4. Comprehensive Ablation Studies
> To rigorously dissect the performance gains of MAPPS and isolate the contributions of each module, we conducted an extensive ablation study across multiple dimensions:
>
> #### A. Ablation of Agent Core Components
> * **Tool Code Generator Ablation:** As the foundational execution engine of MAPPS, the Tool Code Generator cannot be entirely removed, but its core capability is heavily dependent on the surrounding agent feedback loops.
> * **Scientific Mediator Ablation:** Removing the Scientific Mediator disables the automated code error gatekeeping and workflow auditing, forcing the system to either crash upon encountering minor syntax bugs or severely spam the human scientist with constant retry commands.
> * **Workflow Planner Ablation:** Replacing the Workflow Planner with a static template causes a severe performance drop across tasks. Without the dynamic planning layer, the rigid template fails to handle complex physical logic and necessitates more intensive human feedback to salvage the pipeline; for instance, the human must manually intervene to enforce strict stoichiometric ratios rather than just matching element types during retrieval, alongside manually injecting critical validation rules for crystal symmetry and charge balancing.
>
> #### B. Ablation of Other System Components
> * **Without Human Intuition and Feedback:** Without human high-level insights, pure autonomy forces the Workflow Planner to rely on web-scale priors that mistakenly attempt to train generative models from scratch in a tool environment. When restricted to a rigid prototype-design template without dynamic agent optimization or subsequent human feedback, performance drops drastically across benchmarks, yielding Match Rates and mean RMSEs of 58.4% (0.0205) on MP20, 19.1% (0.0388) on MPTS, and 22.3% (0.0425) on the Challenge Set. This underscores that unguided execution fails to maintain robust workflow trajectories.
> * **Without Retrieval Source:** We emphasize that a purely non-retrieval, ab initio setup is not physically practical. Prior materials science studies have shown that blind structure generation is fundamentally unreliable and computationally expensive; for example, even predicting a simple $\text{Al}_2\text{O}_3$ structure may require evaluating hundreds of random configurations. In this sense, existing generative models such as CDVAE also rely on structural priors learned from training data, although these priors are implicit. By contrast, MAPPS makes this reuse explicit through tool-based retrieval and adaptation of structural prototypes. This design is therefore not merely a convenience, but a physically necessary mechanism for improving feasible structure generation.
>
> * **Impact of MLFF:** For the CSP task (where table baselines exclude relaxation for a fair baseline comparison), adding CHGNET relaxation yields high sub-angstrom geometric precision, successfully optimizing candidate structures to an exceptionally precise RMSE of 0.017 on MP20, 0.033 on MPTS, and 0.037 on the Challenge Set. Due to the well-known fidelity gap between empirical MLFF potentials and true DFT functional landscapes, the rigid geometric Match Rates shift to 63.30% (MP20), 27.10% (MPTS), and 29.31% (Challenge Set) respectively, which reflects realistic local coordinate minimization. Conversely, for the CSG task, relaxation is embedded strictly as a sparse, high-efficiency feedback loop used only to evaluate a few test cases during code iteration, while the CSD task completely omits downstream relaxation as it focuses purely on property-targeted screening rather than thermodynamic stability.

---

### Review · Reviewer_H6HG · 2026-04-28

**Summary Of Contributions:**

## Strengths

**1. Meaningful problem formulation**

The paper addresses a valuable and timely problem by moving beyond standalone crystal generators toward a **closed-loop scientific system**. Instead of focusing solely on generation, it decomposes materials discovery into an integrated pipeline of **planning, code generation, physics-based tools, and human feedback**. This system-level perspective is important for AI for Science, particularly in understanding how agents can **reliably orchestrate scientific workflows**.

---

**2. Strong experimental coverage**

Compared to typical proof-of-concept agent papers, the experimental evaluation is relatively comprehensive. The authors consider multiple tasks, including:

- crystal generation
- crystal structure prediction (CSP)
- bandgap-constrained generation

On MP-20, MAPPS achieves:

- **34.3% DFT stability rate**
- **24.9% S.U.N. rate**

which outperform baselines such as CDVAE, DiffCSP, FlowMM, and FlowLLM.

For CSP tasks (MP-20, MPTS-52, and challenge sets), MAPPS also reports higher match rates than prior methods.

---

**Audience:**

Yes

**Audience Explanation:**

yes

**Claims And Evidence:**

Yes

**Claims Explanation:**

While the direction is timely and important, I have several concerns regarding attribution of improvements, the strength of autonomy claims, and the clarity of empirical evidence.

---

## Major Concern 1: Unclear Attribution of Performance Gains

The paper reports strong improvements over prior crystal generation methods. However, it is unclear whether these gains arise from:

- the proposed **agentic workflow planning**, or
- the integration of downstream components such as:
    - retrieval from training data
    - MLFF-based relaxation
    - property evaluation tools
    - screening/filtering stages

MAPPS is a composite pipeline, whereas most baselines (e.g., CDVAE, DiffCSP, FlowMM) are pure generative models. This makes the comparison potentially unfair. And no comparison with stronger hybrid baselines (e.g., diffusion + MLFF relaxation)

###

---

## Major Concern 2: Limited Evidence for Autonomous Workflow Design

The paper claims that MAPPS enables **autonomous workflow design**, distinguishing it from prior agent systems.

However, experimental evidence suggests strong dependence on **human intuition**:

- Table 5 shows that **without human intuition, all tested LLMs achieve 0% workflow validity**

This raises concerns about whether the system is truly autonomous.

### Key questions:

1. How much of the performance gain comes from **human-provided workflow guidance**?
2. Are generated workflows **genuinely novel**, or largely prompt-engineered templates?
3. How sensitive are results to **prompt design and human intervention**?

###

---

## Major Concern 3: Overstated Performance Claims

The paper claims a **"five-fold improvement in stability, uniqueness, and novelty rates"**, which may be misleading.

Given that MAPPS includes:

- retrieval
- relaxation
- filtering
- iterative refinement

these gains likely reflect **pipeline-level improvements**, rather than a fundamental advance in generative modeling.

###

---

## Minor Concern: Reproducibility and Transparency

The system relies on multiple interacting components:

- LLMs (via API)
- external physics tools
- retrieval databases

However, important implementation details are missing as follows:

- prompt templates
- failure/retry rates
- number of LLM/tool calls
- computational budget (e.g., DFT usage/MLFF usage)

###

**Requested Changes:**

To strengthen the paper and better support its claims, I recommend the following:

### 1. Clear Attribution via Ablation Studies

The paper would benefit from a more rigorous decomposition of the system to isolate the contribution of each component. In particular, please consider adding ablations such as:

- **MAPPS without MLFF relaxation**
- **MAPPS without retrieval**
- **MAPPS with fixed (non-learned) workflows**

In addition, it would be important to include comparisons against **stronger hybrid baselines**, such as diffusion-based models augmented with:

- MLFF-based relaxation
- DFT-informed supervision or filtering

 Without these analyses, it is difficult to determine whether the observed performance gains are primarily due to the **agent planning component** or downstream pipeline elements.

---

### 2. Stronger Evidence for Workflow Autonomy

To substantiate the claim of autonomous workflow design, the paper should provide more concrete evidence and analysis. Specifically:

- Provide **concrete examples** of:
    - generated workflows
    - generated code
- Compare against **fixed, human-designed workflows**
- Evaluate the **generalization** of generated workflows across different tasks

These analyses would help clarify whether the system is truly learning to design workflows, or primarily relying on structured prompts and human guidance.

Based on current evidence, MAPPS appears closer to a **human-guided orchestration system** rather than a fully autonomous planner.

---

> ### Author Response · Authors · 2026-05-21
>
> We thank the reviewer for the thoughtful and constructive comments. We appreciate the recognition of MAPPS’s system-level perspective and experimental coverage. Below, we address the concerns regarding performance attribution, workflow autonomy, performance claims, and reproducibility.
>
> ### 1. Attribution of Performance Gains
>
> We agree that MAPPS is a composite workflow, and we have revised the manuscript to clarify the role of each component rather than attributing all gains to the planner alone.
>
> * **Retrieval from training data:** Prototype retrieval is physically necessary for practical crystal generation. A purely ab initio setup is highly unreliable and computationally expensive; even simple systems such as $\text{Al}_2\text{O}_3$ may require hundreds of blind configurations. Generative models also rely on structural priors learned from training data, but implicitly, while MAPPS makes this reuse explicit through retrieval and prototype adaptation.
>
> * **MLFF-based relaxation:** MLFF relaxation is used only sparsely in the workflow-design loop, where the agent tests and refines algorithms on a small number of examples before large-scale evaluation. In large-scale crystal generation, both agentic and generative methods generally require MLFF relaxation or more expensive DFT relaxation for physical validation.
>
> * **Property evaluation tools:** These tools are used only for the CSD task, where property-targeted discovery is the objective. Their role is analogous to MLFF energy evaluation in CSG: sparse feedback for designing and adjusting the algorithm, not a universal post-hoc filter.
>
> * **Screening/filtering stages:** MAPPS does not screen generated candidates before evaluation. In CSG, the system generates 1,000 candidates and evaluates all 1,000. We further added a larger-scale experiment with 10,000 generated candidates, achieving a thermodynamic stability rate of 28.8% and a Structurally Unique New materials (SUN) rate of 19.2%. These results show that the improvement does not come from filtering a large candidate pool and reporting only selected samples. Instead, MAPPS guides exploration toward element combinations and structural prototypes that are more likely to yield novel, stable, and unique structures.
>
> ---
>
> ### 2. Evidence for Human-Guided Workflow Design
>
> We agree that MAPPS relies on human-provided scientific priors, which is consistent with our original positioning of MAPPS as a **Level 2 human-guided system**. Human intuition provides the initial scientific direction, while the agent performs dynamic workflow planning, tool execution, code refinement, and scale-up.
>
> To isolate the role of the Workflow Planner, we replaced it with a rigid prototype-design template. Without dynamic planning, performance drops to Match Rates and mean RMSEs of 58.4% (0.0205) on MP20, 19.1% (0.0388) on MPTS, and 22.3% (0.0425) on the Challenge Set. This shows that fixed workflow templates are insufficient for handling diverse crystalline constraints.
>
> We also evaluated fixed human-designed non-agent workflows. A basic retrieval-and-substitution baseline inspired by MatLLMSearch, using identical stoichiometric ratios and element categories, achieves only 3.4% Match Rate on the Challenge Set. A stronger human-engineered version with chemical-family grouping and space-group symmetry preservation improves to 59.0% (0.0208 Å) on MP20, 23.7% (0.0425 Å) on MPTS, and 27.6% (0.0482 Å) on the Challenge Set.
>
> These results show that human priors are important, but they do not fully explain MAPPS’s performance. MAPPS improves over fixed templates by combining human-provided scientific priors with the agent’s internal knowledge and exploratory capability.
>
> ---
>
> ### 3. Clarification of Performance Claims
>
> Building on the component-level clarifications in Concern 1, we further clarify that MAPPS’s gain does not come from directly modeling and sampling the training-data distribution. Instead, MAPPS performs controlled generation by identifying promising structural prototypes, element combinations, and physically meaningful search directions before exploring candidate structures.
>
> This distinction matters because current crystal generative models have limited controllability. Their stability rates may be reasonable, but novelty and uniqueness are harder to control, leading to a large drop from stability rate to the stricter S.U.N. metric. MAPPS reduces this gap by guiding generation toward regions that are more likely to yield stable, unique, and novel materials.

---

> ### Author Response · Authors · 2026-05-21
>
> ---
>
> ### 4. Reproducibility and Transparency
>
> We have added additional implementation details on failure/retry behavior, LLM/tool-call usage, and computational budget.
>
> * **Failure/retry rates:** We provide the code bug rate and clarify whether errors were resolved by self-reflection or required human intervention. In CSP, the observed code bug rate was 8.3%, and syntax/runtime errors were resolved autonomously through the self-reflection loop. The three human interventions were for high-level workflow clarification rather than code debugging. For CSG and CSD, both tasks had a 0% code bug rate and required no additional human intervention during execution.
>
> * **Number of LLM/tool calls:** For CSP, MAPPS used 30 LLM API calls and about 120,000 tokens. For CSG, MAPPS used 12 LLM API calls. For CSD, MAPPS used 8 LLM API calls.
>
> * **Computational budget:** For CSP, MAPPS follows the standard baseline protocol by generating 1 candidate per target with no downstream DFT evaluation. For CSG, MAPPS generated 1,000 candidates and evaluated all 1,000 candidates, with no selective screening before evaluation. MLFF was used only sparsely during the debugging and code-design stage, with 20 MLFF evaluations. For CSD, MAPPS generated 500 property-targeted candidates using Comformer predictors without downstream DFT calculations.
>
> ---
>
> ### 5. Response to Requested Changes
>
> For **clear attribution**, we now separately discuss retrieval, MLFF relaxation, property evaluation, and screening/filtering. We also add fixed-workflow comparisons, including Workflow Planner ablation and two human-designed non-agent retrieval-and-substitution baselines.
>
> For **MAPPS without MLFF relaxation**, we clarify that MLFF is not used as a large-scale post-hoc optimization stage. In CSG, MLFF is used only sparsely during the workflow-design loop, with 20 evaluations for debugging and code refinement before large-scale deployment, while the final evaluation reports all generated candidates without MLFF-based filtering. For CSP, the main baseline comparison follows the standard protocol without downstream relaxation. We additionally report CHGNET-relaxed results as a separate analysis of local geometric refinement. Under CHGNET relaxation, RMSE improves to 0.017 on MP20, 0.033 on MPTS, and 0.037 on the Challenge Set, with corresponding Match Rates of 63.30%, 27.10%, and 29.31%, respectively. Due to the fidelity gap between MLFF potentials and true DFT landscapes, these results primarily reflect local coordinate minimization rather than a change in the workflow itself.
>
> For **stronger hybrid baselines**, we clarify that existing crystal generation baselines are typically evaluated after structural relaxation, either through MLFF-based relaxation or more expensive DFT relaxation. Therefore, our comparison is not against completely unrelaxed generative outputs. We agree that using MLFF feedback to improve generative models through on-policy learning is a feasible and potentially strong direction. However, this is different from our CSG setting, where MLFF is used only sparsely to help the agent design and adjust code, rather than to train or iteratively improve a generative model.
>
> For **fixed human-designed workflows**, we have already added direct comparisons. A basic hard-coded retrieval-and-substitution baseline achieves only 3.4% Match Rate on the Challenge Set, while a stronger human-engineered version with chemical-family grouping and space-group symmetry preservation achieves 59.0% (0.0208 Å) on MP20, 23.7% (0.0425 Å) on MPTS, and 27.6% (0.0482 Å) on the Challenge Set.
>
> For **workflow examples and generated code**, we will include more detailed generated workflows and code examples in the appendix.

---

### Review · Reviewer_S284 · 2026-05-07

**Summary Of Contributions:**

This paper presents MAPPS, a human-guided multi-agent framework for crystal materials discovery. The system has three parts: a Workflow Planner, a Tool Code Generator, and a Scientific Mediator. The planner proposes a short scientific workflow from a high-level task and human intuition. The code generator turns each step into executable Python code and calls tools such as ML force fields, property calculators, and structure analyzers. The mediator coordinates human feedback and error recovery. The authors position MAPPS as a Level 2 autonomy system, which is more flexible than a fixed tool-execution pipeline, but not fully autonomous.

The experiments cover crystal generation, crystal structure prediction, and bandgap-constrained generation. The reported numbers are strong, MAPPS improves the S.U.N. rate on MP-20, performs well on CSP benchmarks, and achieves high condition satisfaction in bandgap-targeted generation. The paper also includes a useful workflow-validity study showing that current LLMs fail at fully unguided workflow planning, while human intuition substantially improves workflow validity.

The main strength is the problem choice. This is a timely question, and the paper is right that many scientific agents are still basically workflow executors. The main weakness is that the current evidence does not cleanly show where the gains come from. MAPPS uses retrieval, ML force-field relaxation, DFT validation, human intuition, and generated code. Any of these could be doing most of the work. The paper needs stronger controls before the agentic-planning claim is convincing.

**Additional Comments:**

This is an interesting paper, and the direction is worth pursuing. I like that the authors do not pretend current LLMs have reached full autonomy. The Level 2 framing is the right one, and the workflow-validity experiment is useful. To be short: promising idea, timely problem, but the main claims need sharper controls and more careful wording before acceptance.

**Audience:**

Yes

**Audience Explanation:**

The topic is clearly relevant to TMLR readers interested in AI for science, LLM agents, materials discovery, and scientific workflow automation. The paper asks a good question: can LLM agents move beyond fixed tool-use pipelines and help plan scientific workflows with human guidance? That question is timely and worth studying.

**Broader Impact Concerns:**

The paper should add a short Broader Impact section.
The main risk is over-trust. MAPPS may produce structures that look validated because the workflow includes LLM planning, ML force fields, DFT, and human feedback, but this does not mean the system is independently discovering materials in the strong sense. The authors should clearly say that MAPPS is a workflow-assistance and hypothesis-generation system, not a replacement for expert validation.

**Claims And Evidence:**

No

**Claims Explanation:**

The paper has promising results, but some of the strongest claims are ahead of the evidence. The results show that MAPPS can be a useful workflow, but they do not yet prove that the LLM planning component is the main driver of the gains.

The headline “five-fold improvement” is a good example. Table 1 supports a roughly five-fold improvement in S.U.N. rate on MP-20 compared with FlowLLM: 24.9% for MAPPS versus 4.92% for FlowLLM. But the wording in the abstract says “a five-fold improvement in stability, uniqueness, and novelty rates,” which reads as if all three components improve five-fold. That is not what the table shows. The DFT stability rate improves from 17.8% to 34.3%, which is meaningful but not five-fold.

The comparison to generative models is also not quite clean. MAPPS uses the dataset as a retrieval database, then applies structure modification, ML force-field relaxation, and DFT-based evaluation. Baselines such as CDVAE, DiffCSP, FlowMM, CrystalTextLLM, and FlowLLM are trained generative models. Comparing the full MAPPS system against those methods is interesting, but it is not an apples-to-apples comparison of generative modeling capability. The paper should compare against a strong non-agent baseline such as “nearest-neighbor retrieval + composition substitution + CHGNet or M3GNet relaxation + energy ranking.”

The autonomy claim also needs tightening. The workflow-validity experiment is actually quite revealing: without human intuition, all tested models get 0% validity across the three tasks; replacing the human mediator with another LLM also fails. That is an important result, but it supports a conservative conclusion: MAPPS is a human-guided workflow assistant, not evidence of near-full autonomous discovery.

**Requested Changes:**

## Critical:
1. The most important missing baseline is a non-agent pipeline that retrieves similar structures, substitutes or modifies compositions, relaxes candidates using CHGNet or M3GNet, and ranks by energy. Without this, it is hard to know whether MAPPS is better because of agentic planning or simply because retrieval plus physics-based relaxation is strong.

2. The paper should isolate the contribution of the Workflow Planner, Tool Code Generator, Scientific Mediator, human intuition, retrieval database, self-reflection, and MLFF relaxation. At minimum, compare fixed human workflow, LLM-planned workflow, no human intuition, no self-reflection, no retrieval, and no relaxation.

3.  The paper should state how many candidates are generated, how many are relaxed, how many are evaluated by DFT, how many API calls are used, and how much human feedback is required. These are not minor details, they define the real cost and autonomy level of the system.

4. The paper should consistently describe MAPPS as a Level 2 human-guided system. The authors’ own results show that unguided workflow planning fails. That finding is valuable, but it should temper the autonomy framing.

## Would strengthen the paper:
1. Report variance across multiple LLM sampling runs and retrieval choices.

2. Add baselines for bandgap-constrained generation, such as random retrieval, nearest-property retrieval, and a MatLLMSearch-style search baseline.

3. Report code-generation success rate, reflection success rate, and average number of retries.

4. Include failure cases where workflows are invalid, relaxation fails, or retrieved prototypes are misleading.

---

> ### Author Response · Authors · 2026-05-21
>
> We thank the reviewer for the constructive feedback, encouraging comments, and insightful questions. We address each of your comments and requested changes below.
>
> ### 1. Comprehensive Ablation Studies
>
> To better isolate the contribution of each module in MAPPS, we conducted ablation studies across both agent core components and system-level components.
>
> #### A. Ablation of Agent Core Components
>
> * **Tool Code Generator Ablation:** The Tool Code Generator is the foundational execution engine of MAPPS and therefore cannot be entirely removed. However, its effectiveness depends strongly on the surrounding planning and feedback loops.
>
> * **Scientific Mediator Ablation:** Removing the Scientific Mediator disables automated code error checking and workflow auditing. As a result, the system either crashes when encountering minor syntax or execution errors, or requires substantially more human intervention for retry and debugging.
>
> * **Workflow Planner Ablation:** Replacing the Workflow Planner with a static template causes a clear performance drop across tasks. Without dynamic planning, the pipeline struggles to handle complex physical constraints, such as enforcing strict stoichiometric ratios during retrieval, crystal symmetry validation, and charge-balance constraints.
>
> #### B. Ablation of Other System Components
>
> * **Without Human Intuition and Feedback:** Without high-level human guidance, the Workflow Planner relies heavily on broad web-scale priors and may generate impractical workflows, such as attempting to train generative models from scratch in a tool environment. Under a rigid prototype-design template without dynamic agent optimization or later human feedback, performance drops to Match Rates and mean RMSEs of 58.4% (0.0205) on MP20, 19.1% (0.0388) on MPTS, and 22.3% (0.0425) on the Challenge Set.
>
> * **Without Retrieval Source:** We emphasize that a purely non-retrieval, ab initio setup is not physically practical. Prior materials science studies have shown that blind structure generation is unreliable and computationally expensive; even predicting a simple $\text{Al}_2\text{O}_3$ structure may require hundreds of random configurations. Generative models such as CDVAE also rely on structural priors learned from training data, although implicitly. MAPPS instead makes this reuse explicit through tool-based retrieval and adaptation of structural prototypes.
>
> * **Impact of MLFF:** For the CSP task, where table baselines exclude relaxation for fair comparison, adding CHGNET relaxation improves local geometric precision, achieving RMSEs of 0.017 on MP20, 0.033 on MPTS, and 0.037 on the Challenge Set. The corresponding Match Rates are 63.30%, 27.10%, and 29.31%, respectively. For CSG, relaxation is used only as a sparse feedback loop during code iteration, while CSD omits downstream relaxation because it focuses on property-targeted screening rather than thermodynamic stability.
>
> ---
>
> ### 2. Concerning the Requested Non-Agent Baseline (Retrieval + Substitution)
>
> We evaluated a human hard-coded, non-agent retrieval-and-substitution baseline inspired by MatLLMSearch. A basic rule-based version, which retrieves prototypes by identical stoichiometric ratios and element categories, is highly brittle and achieves only 3.4% Match Rate on the Challenge Set.
>
> We therefore also tested a stronger human-engineered version with chemical-family grouping and space-group symmetry preservation. This baseline achieves Match Rates and mean RMSEs of 59.0% (0.0208 Å) on MP20, 23.7% (0.0425 Å) on MPTS, and 27.6% (0.0482 Å) on the Challenge Set.
>
> This shows that retrieval plus substitution is useful for familiar structures, but fixed pipelines remain limited under diverse physical constraints.
>
> ---
>
> ### 3. Quantitative Computational Cost
>
> For **CSP**, MAPPS follows the standard baseline protocol by generating 1 candidate per target with no downstream DFT evaluation. It used 30 LLM API calls and about 120,000 tokens. The observed code bug rate was 8.3%, and these syntax/runtime errors were resolved autonomously through the self-reflection loop. Human intervention was required only 3 times, not for code debugging, but to clarify high-level workflow rules: resolving a file-naming convention, expanding group-level chemical similarity constraints, and enforcing symmetry protection on equivalent sites.
>
> For **CSG**, MAPPS generated 1,000 candidates and ran 1,000 validation DFT calculations. The workflow used MLFF pre-relaxation to reduce the cost of direct ab initio DFT search. It required 12 LLM API calls, had a 0% code bug rate, and required 0 human interventions.
>
> For **CSD**, MAPPS generated 500 property-targeted candidates without downstream DFT calculations, using Comformer property predictors instead. This task used 8 LLM API calls, had a 0% code bug rate, and required 0 human interventions.

---

> > ### Author Response · Authors · 2026-05-21
> >
> > ---
> >
> > ### 4. Concerning the System Categorization as a Level 2 Human-Guided System
> >
> > We agree with the reviewer and have revised the manuscript to consistently describe MAPPS as a Level 2 human-guided system. Our ablation results also support this categorization: unguided autonomy and rigid templates perform poorly in complex crystalline spaces, while MAPPS benefits from combining high-level human guidance with dynamic agent-based workflow execution.
> >
> > ---
> >
> > ### 5. Extended Experimental Evaluations and Large-Scale Results
> >
> > We added two new experimental evaluations to further assess MAPPS’s scalability and discovery capability.
> >
> > * **10-Shot Evaluation on Challenge Set:** We evaluated a 10-shot retrieval configuration on the Challenge Set. MAPPS achieves a Match Rate of 37.9%, showing improved structural coverage with an expanded prototype budget.
> >
> > * **Large-Scale Generation and Discovery on MP20:** We conducted a large-scale generation run producing 10,000 candidate structures. Downstream validation shows a thermodynamic stability rate of 28.8% and a Structurally Unique New materials (SUN) rate of 19.2%, supporting MAPPS’s utility for large-scale materials exploration.
> >
> > ---
> >
> > ### 6. Response to Suggestions for Strengthening the Paper
> >
> > We thank the reviewer for these helpful suggestions. In the revision, we will address variance analysis, bandgap-constrained baselines, code-generation success metrics, and failure case studies.

---

### Decision · Action_Editor_Jpx4 · 2026-06-22

**Recommendation:** Accept with minor revision

**Additional Comments:**

All the details provided in the rebuttal should be included in the next version. This include critical ablation study results, comparison with L1 baselines, attributions over the components, and the effect of human guidance. Descriptions on at least the settings and evaluation protocols of these additional experiments should be provided to ensure reproducibility.

**Audience:**

Yes

**Audience Explanation:**

Leveraging AI systems for an autonomous research system is a potentially highly impactful and fast developing direction. The work tries to improve the level of autonomy by introducing a planner agent, and more preferred results are produced. All the three reviewers voted "leaning accept" regarding the inspiration and the better results.

**Claims And Evidence:**

Yes

**Claims Explanation:**

Not all the claims are well supported initially, as pointed out by some reviewers, e.g., being better than a Level 1 agent system, the effect of the three components (especially the Workflow Planner, the key distinction from a Level 1 system), and the effect of human intuition. The support for the claims becomes stronger after the rebuttal period, provided that the results and settings of the L1 baseline comparison and additional ablation studies are properly implemented into the paper.